# REVISITING THE SUPERFICIAL ALIGNMENT HYPOTHESIS

## ABSTRACT

The Superficial Alignment Hypothesis posits that almost all of a language model's abilities and knowledge are learned during pre-training, while post-training is about giving a model the right style and format. We re-examine these claims by empirically studying the scaling behavior of post-training with increasing fine-tuning examples and evaluating them using objective task-specific standardized benchmarks. Through experiments with the Llama-3, Mistral, and Llama-2 model families of multiple sizes, we observe that, similar to the pre-training scaling laws, post-training task performance scales as a power law against the number of finetuning examples. This power law relationship holds across a broad array of capabilities, including mathematical reasoning, coding, instruction following, and multihop-reasoning. In addition, for tasks like math and multihop reasoning, we observe that a handful of examples merely align the model stylistically but do not saturate performance on the benchmarks. Model performance is instead correlated with its reasoning ability and it improves significantly with more examples, illustrating the need for holistic evaluation programs leveraging objective benchmarks in addition to measurement of alignment to human preferences. We also observe that language models are not necessarily limited to using knowledge learned during pre-training. With appropriate post-training, a model's ability to integrate new knowledge greatly improves on downstream tasks like multihop question-answering. Taken together, these results shed new light on the Superficial Alignment Hypothesis, suggesting that it is, at best, an over-simplification.

## 1 INTRODUCTION

Large Language Models (LLMs) based on the Transformer architecture have achieved state-of-the-art performance on tasks that involve instruction following, problem-solving, and reasoning (Vaswani et al., 2017; Achiam et al., 2023; Dubey et al., 2024). The standard pipeline for building LLMs powered applications involves unsupervised training of a model on a giant corpus of data to gain general language understanding capability, referred to as *pre-training* (Radford et al., 2019; Brown et al., 2020). The model is further improved using *post-training*, which involves finetuning it to excel at a particular domain or behave like a helpful chatbot. This process is also referred to as alignment. The predominant way to do this is through Supervised Finetuning (SFT) where the language model is provided with a prompt, and the model is finetuned to respond to the task (Wei et al., 2022). An additional step is Reinforcement Learning through Human Feedback (RLHF) where a model is trained using reinforcement learning to generate human-preferred responses, by being rewarded for good responses and penalized for bad responses (Ouyang et al., 2022).

To achieve the post-training goal of responding appropriately to various user queries, LLMs need to develop several task-specific capabilities, like mathematics, reasoning, utilizing knowledge, and tool use. To teach a model these capabilities, model builders collect human-annotated or synthetically generated data and finetune the model to obtain the desired behavior. Since data collection at scale is labor and cost-intensive, it is essential to understand the qualitative and quantitative value of obtaining additional finetuning data. Studies like LIMA by Zhou et al. (2024) have hypothesized that post-training alignment is all about learning the style and format of the desired behavior. Specifically, it puts forward the Superficial Alignment Hypothesis, whose claims are:

- **C1:** A model's knowledge is learned entirely during pre-training.

- **C2:** Post-training is largely about style and doesn't does not teach a model new capabilities.

- **C3:** A small number of examples can saturate a model's performance for a given task.

However, the experiments from LIMA and follow-up works by Lin et al. (2023) primarily evaluate chatbot style interaction capabilities - tasks that require mostly cosmetic changes to a model's response style. It is unclear how these models improve on task-specific reasoning capabilities during post-training. They are also evaluated using a subjective win-rate comparison, over open-ended prompts. This doesn't provide an objective pattern to analyze model behavior and thus fails to provide useful information about the nature of model performance and dataset size. For researchers and practitioners who finetune LLMs to perform specific tasks, understanding scaling behavior with more data is crucial in aiding data collection and annotation efforts. There is also a need to study if these LLMs are limited to the knowledge acquired during pre-training, or if we can introduce new knowledge and show how to utilize it effectively. So, we design three research questions to better investigate these claims:

1. *How does post-training model performance scale with finetuning dataset size?*

2. *Is the model significantly improving on task-related capabilities or just learning the response style?*

3. *Can a model integrate new knowledge from beyond the pertaining knowledge cutoff?*

In the following section, we summarise the key results from the study, followed by sections that detail the experimental setup, results, and conclusions for each of the research questions outlined above.

## 2 KEY TAKEAWAYS

- Post-training performance on a task has a power law relationship of the form $P \propto D^{1/b}$ with the number of finetuning samples, across models of multiple families and sizes, similar to scaling laws established for pretraining and inference(Kaplan et al., 2020; Brown et al., 2024). (Section 3)

- Evaluating alignment models using win-rates as shown in Zhou et al. (2024) could be misleading for reasoning-based tasks. For instance, LLM-based judges can prefer model generations that exhibit a chatbot-style answer for mathematical questions, even though the model might be poor at mathematical abilities as observed on math benchmarks. (Section 3.3)

- Through extensive error analysis on tasks like math and multihop reasoning, we see that when a model is finetuned for a task, the improvements in task-specific style and formatting saturate in just 100 examples, as hypothesized by the Superficial Alignment Hypothesis. However, the model's performance on the task is directly correlated with its improvements in reasoning ability, which improves notably during post-training with more finetuning examples. (Section 4)

- Post-training a model for reasoning can also help a model integrate knowledge beyond its pre-training knowledge cutoff. Compared to pre-trained models, finetuned models learn and use new knowledge on downstream tasks effectively. (Section 5)

These experiments help frame the Superficial Alignment Hypothesis in a new light. Most importantly, the focus on post-training should not be entirely on stylistic alignment, but also on measuring downstream task metrics. When seen through this lens, we see that after finetuning on a few high-quality examples, LLMs behave in the right style and format, especially when evaluated through subjective techniques like win rate. However, this doesn't necessarily warrant the conclusion that the model has been aligned for the task. When evaluated for their objective task-specific performance, we see that the models do improve significantly with additional data on many tasks during post-training over their pre-trained counterparts. In addition, these improvements are primarily driven by improvements in reasoning and analytical abilities during finetuning. Good post-training is also an effective way for LLMs to learn and integrate new knowledge from beyond their knowledge cut-off.

Table 1: Experiment training and test benchmark details

| Task | Test Benchmark | Train Dataset | # train examples |
|------|---------------|---------------|------------------|
| Math | GSM8k Test | GSM8k Train | 7,500 |
| Multihop QnA | SubQA Test | SubQA Train | 2,700 |
| Coding | HumanEval+ | StarCoder Self-Align Train | 10,000 |
| Instruction Following | IFEval | Conifer Hard Messages | 5,000 |
| Instruction Following | IFEval | Dolly15k | 15,000 |

## 3 POST-TRAINING DATA SCALING

*How does post-training model performance scale with finetuning dataset size?*

Following prior art (Zhou et al., 2024; Lin et al., 2023), we interpret pre-training as training a model using a large corpus of text. The primary objective is to gain general-purpose language understanding and is mainly optimized using negative log-likelihood next token prediction loss on a large corpus. In contrast, post-training is the process of finetuning a language model to improve on specific capabilities that are important and relevant to humans - instruction following, math, coding, etc. These are primarily optimized by evaluating through task-specific benchmarks. Although this distinction is not always clear because of intermediate training stages, the simplified setting allows us to thoroughly evaluate the Superficial Alignment Hypothesis and understand the value of finetuning data without many confounding variables.

The primary implication of the Superficial Alignment Hypothesis is that pre-training is all that matters, and with a rather small set of examples, we can align a model during post-training. However, this is a broad claim that is supported by a limited set of chatbot-style experiments. Post-training a model involves finetuning for instruction following, problem-solving, and coding, and unlike chatbot-style dialogue whose evaluation is subjective and comparative, these capabilities can be judged using standardized benchmarks. For researchers and model builders who aim to improve performance on such tasks, it is important to understand performance scaling on such benchmarks with increasing finetuning data.

### 3.1 EXPERIMENT DESIGN

We look at four tasks (training datasets used are in parenthesis) - mathematical problem solving (GSM8k (Cobbe et al., 2021)), instruction following ((Sun et al., 2024; Conover et al., 2023)), coding (StarCoder Self-Align [1]) and multihop question answering (SubQA (Tang et al., 2021)). Starting from the base model and finetuning with increasing dataset size, we observe how performance scales during finetuning. For each task, we ran evaluations using a standard framework where available (Gao et al., 2024; Liu et al., 2023; Zhou et al., 2023). More details about the training data are available in Table 1 and additional dataset construction details can be found in Appendix A.3.

For our experiments, we finetune Llama-3, Llama-2 and Mistral model families on these tasks (Dubey et al., 2024; Touvron et al., 2023; Jiang et al., 2023). We chose base models because it is likely that instruct models are already extensively finetuned for these tasks.

We first finetuned the smallest (sub-10 Billion parameter) models from these model families, with the dataset splits of 0, 100, 500, 1000, 5000, and 10000 examples until the training dataset was exhausted. To study the effects of model parameter size, we also scale up the models in the Llama-2 and Llama-3 families from 7B to 70B parameters - a 10X increase. Since full-parameter fine-tuning of 70B models on a multi-node GPU cluster is resource-intensive, we limited the number of training runs for the 70B model sizes.

To ensure fairness, every model in a model family was trained with the same set of hyperparameters for a given task and dataset size, for 3 epochs over the base model with the default chat template from HuggingFace. More details about the hyperparameters for training and inference can be found in A.7.

---

[1] https://huggingface.co/datasets/bigcode/self-oss-instruct-sc2-exec-filter-50k

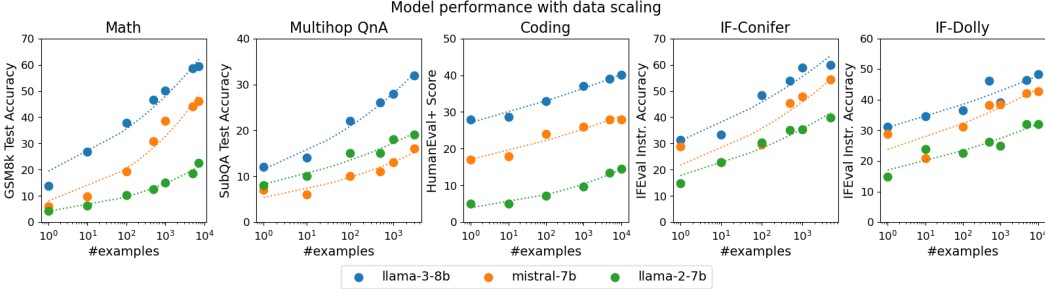

Figure 1: Performance improvements as finetuning data is scaled up, for models in the sub-10 Billion parameter range. The points are fitted with a power law curve of the form $P \propto D^{1/b}$. Model performance consistently scales in a power law fashion, across model families.

## 3.2 RESULTS

**Model performance for a task follows a power-law relationship with finetuning data.** Figure 1 shows the performance scaling with increasing finetuning data for the smallest, sub-10 Billion parameter models, with the power-law fit line. Task accuracy $P$ closely follows a power-law of the form $P \propto D^{1/b}$ with the number of finetuning examples $D$, for all the models on all the tasks. This power-law relationship is in line with several other empirical scaling laws of LLMs with data size during pre-training, quantization, and inference (Kaplan et al., 2020; Michaud et al., 2024; Brown et al., 2024). The coefficients for the power law curves are in Appendix A.8.

In addition, model improvement curves do not cross each other. Better base models for a task are consistently better during post-training as well. This is in line with other works that relate pertaining performance with downstream task performance (Zhang et al., 2024a). Performance scaling with data is also more consistent and predictable on reasoning-centric tasks like Math, Multihop QnA, and Coding, as opposed to subjective tasks like Instruction Following. We also perform additional ablations on scaling curves with different dataset quality for Instruction Following in Appendix A.3.3, and evaluations on the GSM1k benchmark (Zhang et al., 2024b) for math to check for dataset contamination, in Appendix A.1.

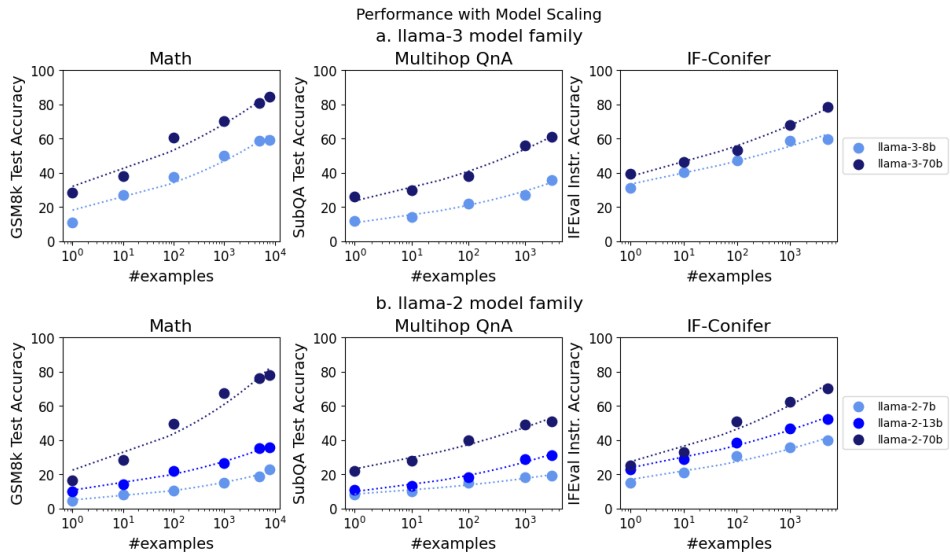

Figure 2: Performance scaling curves with increasing model size for models in the same family.

**Larger models in a model family learn better than smaller models with more data.** Figure 2 shows results for model scaling. All models of the family follow the same power law distribu-

tion. As expected, larger models consistently outperform smaller models for the same training data. However, the improvement is not always just an upward parallel shift. The performance of larger models curves upwards with more training datasets, indicating an increasing rate of improvements with additional data. The coefficients for the power law curves are in Appendix A.8.

Putting these results together, we see that larger and better base models scale even better during post-training. This highlights the role of pre-training in preparing a model to learn better during post-training.

## 3.3 General Purpose Alignment vs Task Specific Finetuning

The LIMA paper also introduced the LIMA dataset, a collection of 1,000 carefully curated prompts that was intended to align a pre-trained model to be on par with state-of-the-art post-trained models. It is based on the hypothesis that the model has inherently learned most of its capabilities during pre-training and thus, the model just needs to adopt a stylistic format to answer questions.

In this section, we finetune the Llama-3 8b model with the LIMA dataset, using the same training setup as the rest of the datasets. We call this model *LIMA-1k*. We also finetune a pre-trained model with 1,000 examples specific to a task - GSM8k for math and SubQA for multihop reasoning. This ensures that the LIMA and task-specific models are trained with a similar data "budget". We call models finetuned with task-specific datasets *Task-1k*. We then evaluate the performance of the chat-bot style aligned LIMA-1k against task-specific fine-tuned Task-1k model on the task-specific benchmarks.

Table 2: Comparision of task-specific finetuning v/s LIMA style stylistic alignment.

| | Task Accuracy | | Win Rate | | |
| --- | --- | --- | --- | --- | --- |
| | LIMA-1k | Task-1k | LIMA-1k | Task-1k | Neither |
| **Math** (GSM8k Test) | 14.7% | **46.5%** | **84.4%** | 0.24% | 14.2% |
| **Multihop QnA** (SubQA Test) | 21% | **36%** | 20% | **57%** | 23% |

**Task specific finetuning largely outperforms stylistic finetuning when evaluated objectively.** From the results in Table 2, we observe the marked difference that domain-specific data and post-training make against just chat-bot style alignment. For the same number of examples, domain-specific post-training greatly outperforms general-purpose chatbot style alignment. In addition, since several works, including LIMA, report win-rate against other models as a performance metric, we also calculate the win-rate of the responses from *LIMA-1k* and *Task-1k* models on the GSM8k and SubQA test set prompts. We use the same prompt from LIMA to judge wins between responses and use GPT-4o (OpenAI, 2024) to predict wins. As seen in Table 2, the win-rate metric is an unreliable indicator of a model's accuracy on a specific task. For instance, for the GSM8k Math task, Even though the *LIMA-1k* model generates a significant number of incorrect responses, it gets a higher win-rate than the task-specific fine-tuned model. This highlights the need for task-specific objective evaluations in addition to comparative win-rate metrics in foundation model evaluation programs.

## 4 Learning Reasoning and Style

*Is the model significantly improving on task-related capabilities or just learning the response style?*

In this section, we investigate what is driving the improvements in these tasks with more data, specifically aiming to delineate improvements in style/formatting versus improvements in reasoning. We do this by evaluating the generations of models with different finetuning levels.

### 4.1 Experiment Design

We evaluate finetuned model generations over math (GSM8k dataset) and multihop QnA (SubQA dataset). We took the Llama-3 8b base model as well as fine-tuned models using 100, 1000, and full training splits of the two datasets.

Both GSM8k and SubQA responses use the Socratic Method of generating subquestions to arrive at the final answer. So, the model is finetuned to follow this specific style of generating a subquestion-answer reasoning chain, a delimiter, followed by the final answer. Examples of expected model response styles and formats are in Appendix A.3.1 and A.3.2.

We then collect all the *incorrect responses* from these models on the test split and annotated them using GPT-4o (OpenAI, 2024). If the responses fail to stick to the previously specified format, it is annotated as *Incorrect Formatting*. If the responses contain an error in their subquestion-answer reasoning steps, we annotate it as *Incorrect Reasoning*. For math, we also check for *Incorrect Arithmetic Calculations*, since they are a major source of model errors. Each error category was evaluated independently for a response, using a tailored prompt. So, an error response can belong to multiple categories. More details about the prompt used for this categorization are in Appendix A.4.

## 4.2 RESULTS

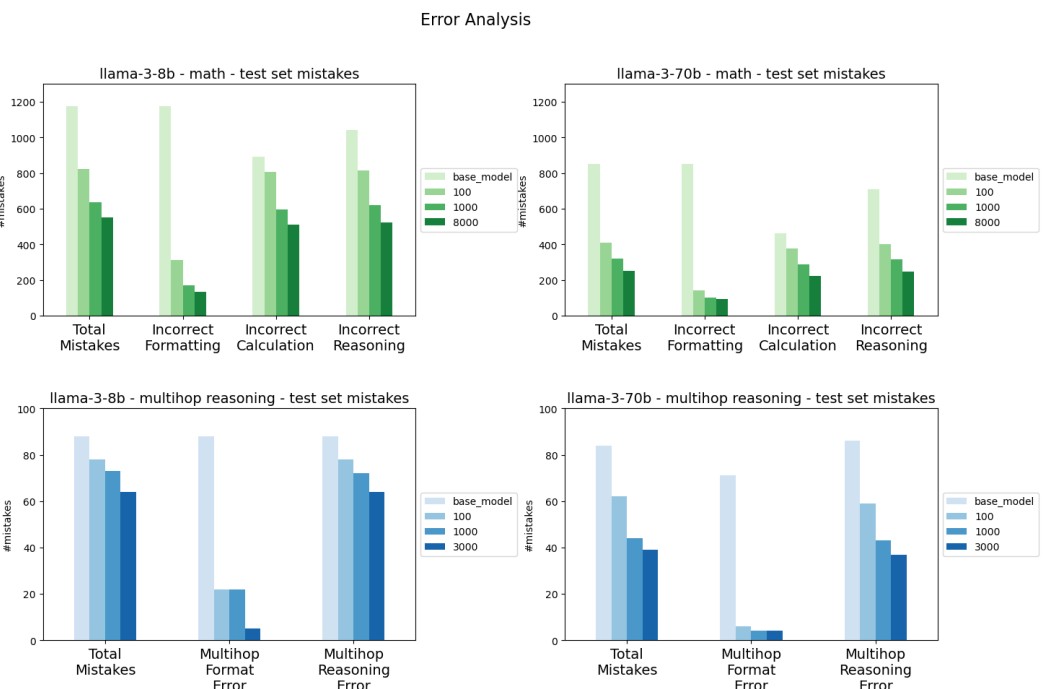

Figure 3: Breakdown of error responses by models finetuned with datasets of increasing data scales. The first group in each chart shows the Total Mistakes made on the test set by the models. Each error response is then independently evaluated for the different mistake types and thus can belong to multiple error types. There is a clear trend of models saturating on style and formatting improvements with just a few examples. However, reasoning and arithmetic errors continue to get better.

**Style and formatting improvements saturate quickly.** From Figure 3 we see that the models get better at style and format errors with just 100 examples. If one just takes a passing look at the responses from this model, they could incorrectly conclude that the model is "aligned" to answer math or multi-hop questions. However, all of these responses are still incorrect for the task for which we finetune the model.

**Reasoning performance continues to improve with more data.** Models continue to get better at reasoning and question understanding with more examples. The total number of mistakes a model makes highly correlates with reasoning errors ($r^2$ value of 0.98 for math and 0.99 for multihop QnA on Llama-3 8B) as opposed to total mistakes and formatting errors ($r^2$ value of 0.93 for math and 0.83 for multihop QnA). It also signifies that a model's capabilities are not entirely learned during pre-training, because models can significantly improve their reasoning, or learn to apply it effectively, during post-training. This leads us to the idea that the superficial alignment hypothesis could

**Event: Miley Cyrus's 'Flowers' was the most streamed song of 2023 on Spotify, whereas Morgan Wallen's 'Last Night' was the most streamed song on Apple Music**

**Direct Question: Which artist had the most streamed song of 2023 on Spotify?**

**Answer: Miley Cyrus**

**Multihop Reasoning Question: Who is the father of the artist with the most streamed song of 2023 on Spotify?**

**Answer: Billy Ray Cyrus**

Figure 4: Example of an Event and Question Pairs from the curated Facts100 dataset.

be limited in scope to improvements on style-and-formatting alignment tasks. It doesn't accurately characterize the improvements in capabilities that post-training is more effective at.

## 5   LEARNING NEW KNOWLEDGE

*Can a model integrate new knowledge from beyond the pertaining knowledge cutoff?*

In this section, we examine how post-training can help LLMs learn new knowledge after the pretaining knowledge cutoff, and more importantly, use it correctly on downstream tasks.

### 5.1   EXPERIMENT DESIGN

We first created *Facts100*, a hand-curated dataset of 100 news events that occurred after March 2023, the knowledge cutoff of the Llama-3-8b base model. The news events are from across the world and cover domains such as entertainment, sports, business, politics and science. We then created two questions for each event, as shown in example 4:

- **Direct Question**: A single-hop direct question related to the news event and its entity.
- **Multihop Question**: A multistep reasoning style question that first requires recalling what happened in the event in the first step *and using it* to answer in the second step. This checks if the model learns how to use the learned knowledge in the right way

Table 3: Accuracy of the models on the Facts100 dataset before introducing new knowledge.

|                   | Direct Qn | Multihop Qn |
|-------------------|-----------|-------------|
| Base Model        | 8         | 12          |
| Post-trained Model| 32        | 27          |

To finetune and evaluate models on this task, we adopt the same task setting as the multihop QnA from the previous section. We evaluated the Llama-3 8B pre-trained model, as well as a multihop reasoning post-trained model. The post-trained model was finetuned on SubQARecall, a modification of the multihop reasoning QnA dataset SubQA from the previous section, augmented to recall the relevant event first before answering the question (more details in Appendix A.3.2. This post-training is meant to impart reasoning ability to the model while maintaining the same knowledge cutoff, since the SubQA dataset doesn't contain any new information after the cutoff. [2] As seen in Table 3, both the models perform poorly on the Facts100 questions, since they have almost no knowledge of most of the new events that are crucial to answer these questions.

---

[2]We didn't use the Llama-3 8B instruct model on the new facts because it performed poorly. This is because it is strongly aligned to refuse to answer questions beyond its finetuning cutoff data. Attempts to get the model to overcome this behavior through finetuning (ex: increasing learning rate) led to behavior degradation. Although unlearning methods can help undo this behavior, it is out of the scope of this study.

## 5.2 INTRODUCING NEW KNOWLEDGE

There are two primary ways to introduce new knowledge to a model - finetuning on the new events or during inference as part of the prompt. The latter is a simplified Retrieval Augmented Generation (RAG) setup Lewis et al. (2020). We investigate the role of post-training in both of these cases.

**Event SFT:** To train the model to learn this new information, we finetune a model on the Direct Question as the prompt and the Event + Answer as the response. The format is the same as the SubQARecall to keep the data in-distribution.

**Event RAG-Oracle:** In this method, we simulate a RAG setup in which knowledge is added in the prompt during inference instead of training on it. To isolate the errors introduced by the retriever component of RAG, we directly add the corresponding event in the prompt, emulating a perfect retriever i.e. an Oracle.

Table 4: Accuracy of the models on the Facst100 dataset with new knowledge, introduced through Event SFT and RAG-Oracle.

| | Event SFT | | Event RAG-Oracle | |
| --- | --- | --- | --- | --- |
| | Direct Qn | Multihop Qn | Direct Qn | Multihop Qn |
| Base Model | 65 | 37 | 49 | 34 |
| **Post-trained Model** | **81** | **55** | **86** | **71** |

### 5.2.1 RESULTS

**Post-training a model for reasoning helps models learn and integrate new knowledge better.** As seen in Table 4, models post-trained for reasoning are significantly better at learning new knowledge (Direct Question) as well as integrating the new knowledge (Multihop Reasoning Question). However, SFT or RAG on the pre-trained model fails to show the same improvement on the harder multi-hop questions. Note that this answer is just one hop from the answer to the direct question but requires it to reason through the steps. This shows that the model can't correctly use this new information in the right way on reasoning tasks.

However, if the model is first post-trained to do reasoning, it gets better at absorbing new information and using it in multihop reasoning tasks. This post-training was done on data from before the knowledge cutoff. Such post-training led to a marked difference in both SFT and RAG-based methods for introducing new knowledge.

**Models hallucination is mitigated by post-training for reasoning, but is not eliminated.** Several studies have shown that LLMs hallucinate when introduced to new knowledge. Since all of the models in our experiments are finetuned to recall the event first and generate the answer based on it, we can easily check for hallucination in the recalled event. We also analyze the subsequent reasoning chain for reasoning errors. Both of these are done using GPT-4o and the prompt used is shown in the Appendix A.3.4.

From Figure 5, we see that hallucinations about the event are a major factor in erroneous model responses. Even after introducing new knowledge, base models hallucinate significantly. This is in line with other works that demonstrate that finetuning models with new knowledge can lead to hallucinations (Gekhman et al., 2024).

However, models post-trained for reasoning make a marked improvement in both hallucination and reasoning errors. Although hallucination is not completely mitigated, the true value of finetuning is in preparing the model to reason with the new knowledge it receives. In addition, RAG-based methods are better than SFT for introducing new knowledge because the relevant information being in the context is a lot more useful. We also do a variety of ablation studies to compare against continual pre-training and LIMA-based finetuning, which are detailed in the Appendix A.2.

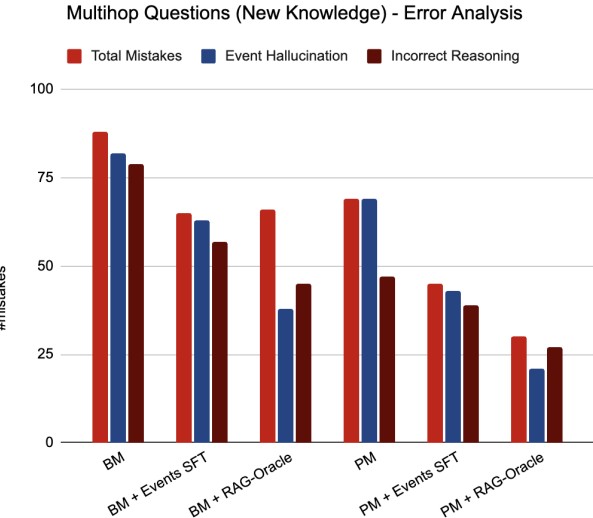

Figure 5: Error Analysis on the New Fact Multihop Questions after fine-tuning. **BM** stands for the pre-trained base model and **PM** for the multihop reasoning post-trained model.

## 6 RELATED WORK

**Language Model Alignment:** LLMs are first pretrained to gain general language understanding and world modeling, which is then followed by an alignment phase that involves SFT and RLHF to appropriately respond to user's prompts (Ouyang et al., 2022; Wei et al., 2021; Dubey et al., 2024). Several recent works laid out their hypothesis on whether alignment is more about learning style and might not even be necessary because of In-Context Learning (Zhou et al., 2024; Lin et al., 2023). More recent works by Zhao et al. (2024) show that instruction tuning on better base models can outperform In Context Learning.

**Scaling Laws:** Several works like Hoffmann et al. (2022); Kaplan et al. (2020); Bahri et al. (2024); Michaud et al. (2024) have studied and developed scaling laws for LLM Pretraining in terms of the dataset token count and the model parameter count against the cross-entropy loss. However, Schaeffer et al. (2024) show that it is hard to predict scaling laws for downstream task performance because of constraints in the action space. Isik et al. (2024); Hernandez et al. (2021) derive scaling laws for finetuning as a function of pretraining data and transfer learning respectively, and not post-training or instruction following. However, researchers and model builders improve model performance by collecting individual prompt-response examples rather than collective dataset tokens (Dubey et al., 2024). Other works like Jiang et al. (2024); Li et al. (2024) also shed light on the performance of models with varying finetuning dataset sizes in specific domains and models.

## 7 CONCLUSIONS AND FUTURE WORK

**Conclusions:** LLM post-training is a complex endeavor that involves improvements to instruction following, stylistic formatting, reasoning abilities, and general alignment to human preferences. LLMs can imitate the required style with "superficial" finetuning using a handful of examples, leading to the Superficial Alignment Hypothesis. However, a solely stylistic evaluation fails to characterize the many aspects of reasoning and task-specific capabilities that are key goals of finetuning. In fact, task-specific skills & reasoning significantly improve after post-training with more examples compared to the pre-trained model. These improvements closely follow a power law in our experiments with the number of finetuning examples across multiple model families and sizes. We also see that these improvements are driven by the model's reasoning ability during generation, and are not limited to the model's alignment to formatting or style. In addition, we see that the win rate against other models can be a misleading metric to measure tasks that require complex reasoning,

signaling the need for holistic evaluation programs leveraging standardized, objective benchmarks, in addition to measurement of alignment to human preferences.

We also observe that good post-training can help LLMs overcome problems associated with knowledge cutoff, by enabling them to better utilize knowledge from beyond the pre-training corpus either via further finetuning or RAG. These results put together highlight the qualitative and quantitative characteristics of post-training, and the role of data scaling in this.

**Limitations and future work:** In this work we showed the performance improvement of a model on a task when it is finetuned with increasing task-specific data. However, frontier LLMs are trained to excel at multiple tasks, and we don't thoroughly understand how finetuning for one task or domain affects the performance on others. A big open question would be investigating how to take advantage of this scaling behavior while preventing model degradation on existing capabilities. Similarly, we showed how models can learn new knowledge beyond their pre-training data cutoff, but the issue of hallucination isn't solved. Further research in effectively introducing new knowledge, like continual learning methods during post-training can shed light on this.

In this work, we also limited the scope to supervised finetuning, on tasks that involve text generation. However, the implications from this opens up several interesting directions to explore further. Most notably, LLM post-training involves RLHF after supervised finetuning, and it would be interesting to see how RLHF can contribute to these improvements and how it scales with more data.

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

# A APPENDIX

## A.1 MATH CONTAMINATION ABLATION

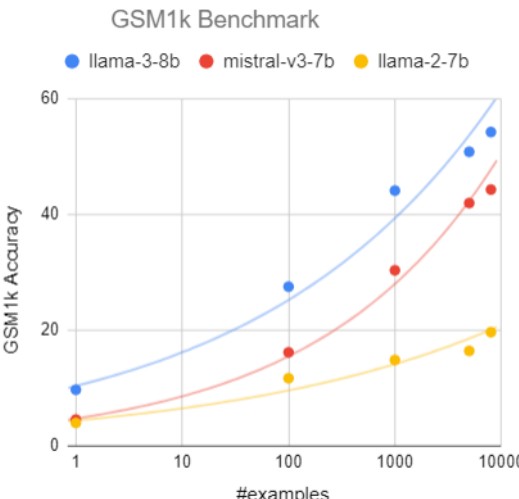

Figure 6: GSM1k benchmark

Since there is a risk that the model might be contaminated with the GSM8k test set during pre-training, we cross-verified the GSM8k-trained Llama-3-8b models with the GSM1k private leaderboard. We observe a similar scaling trend on the new benchmark as well.

## A.2 ABLATION STUDIES FOR INTRODUCING NEW KNOWLEDGE

We also try various ways of introducing new knowledge into the model.

**Continued Pretraining:** The base/posttrained model is first fine-tuned directly on the new events for 3 epochs with the same batch size. The idea is to continue pertaining on the new knowledge corpus.

**LIMA:** We also tried fine-tuning the base model with the LIMA dataset first to make it a general purpose Question-Answering model and provide the required new event as part of the prompt during evaluation, through RAG.

| Method | Direct Qn | Multihop Qn |
|---|---|---|
| Base Model + Continued Pretraining | 35 | 16 |
| Base Model + LIMA + RAG | 58 | 48 |
| Base Model + SFT + RAG | 100 | 51 |
| Posttrained Model w/ Continued Pretraining | 34 | 39 |
| Posttrained Model w/ SFT + RAG | 100 | 71 |

Table 5: Ablation study of different ways to introduce new knowledge.

From table 5, we see that continuing pretraining with new knowledge on the base model or LIMA-style fine-tuning is not effective at introducing new knowledge.

Note that although SFT + RAG models have a perfect score on Direct Question, it is because the SFT models are finetuned on Direct Question-Answer in the first place and is thus, a result of memorization.

## A.3 Dataset Creation and Formatting

### A.3.1 GSM8k Dataset

We chose the Socratic split of the GSM8k dataset since it contains well-developed subquestion-answer steps. This makes it easy to evaluate finetuned model responses for following this format, as well as evaluating the reasoning itself.

An example question-answer pair is given below:

> **System Message:** You are an expert in mathematics. Solve the following math problem
> **Prompt:**
> Question: Natalia sold clips to 48 of her friends in April, and then she sold half as many clips in May. How many clips did Natalia sell altogether in April and May?
> Answer:
> **Response:**
> How many clips did Natalia sell in May? ** Natalia sold 48/2 = <<48/2=24>>24 clips in May.
> How many clips did Natalia sell altogether in April and May? ** Natalia sold 48+24 = <<48+24=72>>72 clips altogether in April and May.
> #### 72

### A.3.2 SubQA Dataset

We modify the SubQA dataset to build a multi-hop reasoning dataset. The original SubQA dataset contains a select 1,000 subset of HotpotQA bridge-style questions, and two human-verified sub-questions for it. For each of the thousand questions, we construct a multi-hop reasoning chain by chaining together the two subquestions, as shown below.

> **Question**: What government position was held by the woman who portrayed Corliss Archer in the film Kiss and Tell?
> **Answer**: Chief of Protocol

Subquestion 1:

> **Question**: Which woman portrayed Corliss Archer in the film Kiss and Tell?
> **Answer**: Shirley Temple

Subquestion 2:

> **Question**: What government position was held by Shirley Temple?
> **Answer**: Chief of Protocol

Generated multi-hop question and reasoning-based answer that the model is fine-tuned on along with the system message:

> **System Message**: You are a helpful assistant. If you are asked a question, answer it carefully and if you are given an instruction, follow it thoroughly. If the question contains some context, use it to answer the question or follow the instruction
> **Prompt**: What government position was held by the woman who portrayed Corliss Archer in the film Kiss and Tell?
> **Response**: What government position was held by the woman who portrayed Corliss Archer in the film Kiss and Tell?
> Let's answer this step-by-step
> Which woman portrayed Corliss Archer in the film Kiss and Tell? → Shirley Temple
> What government position was held by Shirley Temple? → Chief of Staff
> The final answer is Chief of Staff

The SubQARecall version of the dataset was a simple modification of the response format, in which we added an event recall step first, which can be used to evaluate hallucinations. The event annotation was done using GPT-4o using the steps from the response. An example response format for this modified dataset is given below:

> **System Message**: You are a helpful assistant. If you are asked a question, answer it carefully and if you are given an instruction, follow it thoroughly. If the question contains some context, use it to answer the question or follow the instruction
> **Prompt**:
> What government position was held by the woman who portrayed Corliss Archer in the film Kiss and Tell?
> **Response**:
> Event Recall: Shirley Temple played the role of Corliss Archer in the film Kiss and Tell and also held the position of Chief of Staff.
> What government position was held by the woman who portrayed Corliss Archer in the film Kiss and Tell?
> Let's answer this step-by-step
> Which woman portrayed Corliss Archer in the film Kiss and Tell? → Shirley Temple
> What government position was held by Shirley Temple? → Chief of Staff
> The final answer is Chief of Staff

For evaluation, simple string matching ignores answers that are correct but differently worded. It also introduces false positives if the model just contains the final answer as part of its response instead of actually giving it as the answer. So, we use GPT-4o to evaluate the model response and provide the question, the correct answer, and the model-generated answer.

The evaluation prompt for this task is given below

> **System Message**: You are grading responses to a question. You are given a question, response and the answer key.
> **Prompt**: Check if the response to the question is correct, as inferred by the answer key. Here are some guidelines:
> 1. Make sure that the response is actually answering the question and not just writing things about the question.
> 2. If the response is just listing things or spewing exposition related to the question and not actually answering it, it is incorrect. It must be a direct or a multistep reasoning answer to the question.
> 3. The final need not exactly match the answer key, but it should be close enough to be considered correct.
> Return "No Error" if the response is correct and "Error" if it is incorrect and nothing else.
> Now grade this:
> Question - {question}
> Answer - {answer}
> response - {response}

A random subset of 100 multihop questions were held out for evaluation and the rest are used for training.

With this, we get a training dataset of 2,700 questions consisting of 1800 one-hop questions and 900 multi-hop questions. The evaluation set consists of the 100 held-out multihop questions. Since the models are fine-tuned specifically to follow the format of subquestion-answer reasoning steps, we can easily evaluate if they follow the expected reasoning format, as well as evaluate the correctness of the reasoning steps themselves.

### A.3.3 CONIFER AND DOLLY DATASET

The Conifer dataset has multiturn instruction following prompt-response dialogue examples and at each turn, the prompts gets harder, with more constraints and better responses.

For the Instruction Following Tasks, Dolly is of lower quality for the IFEval benchmark because the prompts are very different from the benchmark prompts and do not include constraints and directions. Conifer is a better dataset as it is targeted for constrained instruction following, similar to the IFEval prompts.

An example prompt-response pair from Conifer is given below:

> **System Message**: You are a helpful assistant. If you are asked a question, answer it carefully and if you are given an instruction, follow it thoroughly. If the question contains some context, use it to answer the question or follow the instruction
> **Prompt**:
> Can you provide examples of effective strategies for engaging senior partners in the investment sector through cold-calling? Ensure to cover:
> 1. Across industries such as Private Equity and Venture Capital;
> 2. When communicating with C-suite executives and Managing Directors;
> 3. A concise and to-the-point, data-driven approach;
> 4. Objectives include pitching a new investment opportunity and building a relationship;
> 5. Adherence to financial promotion rules;
> 6. Consideration of time zone differences and avoidance of holidays;
> 7. Demonstrating a thorough understanding of the partner's portfolio and recent industry trends;
> 8. Provide a bulleted list.
> **Response**:
> Effective strategies for engaging senior partners in the investment sector through cold-calling include:
> - Tailoring the pitch to the specific industry, whether it's Private Equity or Venture Capital, highlighting relevant trends and opportunities.
> - Addressing C-suite executives and Managing Directors with respect and understanding of their time constraints, ensuring brevity and relevance in the conversation.
> - Utilizing a data-driven approach to substantiate the investment opportunity, presenting clear metrics and potential ROI.
> - Clearly stating the purpose of the call, whether it's to introduce a new investment opportunity or to initiate a relationship for future collaboration.
> - Ensuring compliance with financial promotion rules by providing accurate information and not making misleading claims.
> - Scheduling calls considering the recipient's time zone and avoiding holidays to respect their personal time.
> - Demonstrating knowledge of the partner's current portfolio and aligning the pitch with their investment strategy and recent industry trends.

An example prompt-response pair from the Dolly dataset is given below:

> **System Message**: You are a helpful assistant. If you are asked a question, answer it carefully and if you are given an instruction, follow it thoroughly. If the question contains some context, use it to answer the question or follow the instruction
> **Prompt**:
> How do I start running?
> **Response:**
> Make sure you get comfortable running shoes and attire. Start with achievable goal in mind like a 5K race. If you never ran before, start gradually from a walk, to brisk walk, light jog aiming for 15-30mins initially. Slowly increase your running time and distance as your fitness level improves. One of the most important things is cool down and gentle stretching. Always listen to your body, and take rest days when needed to prevent injury.

We also do an ablation study of fine-tuning Llama-3-8b against Conifer's hard and easy split as well. We take dialogue examples with at least 4 turns and consider all first turns to create the ConiferEasy dataset and all last turns to create the ConiferHard dataset. Note that both of these are now single-turn prompt-response datasets. We also compare these models against the Dolly15k dataset. As

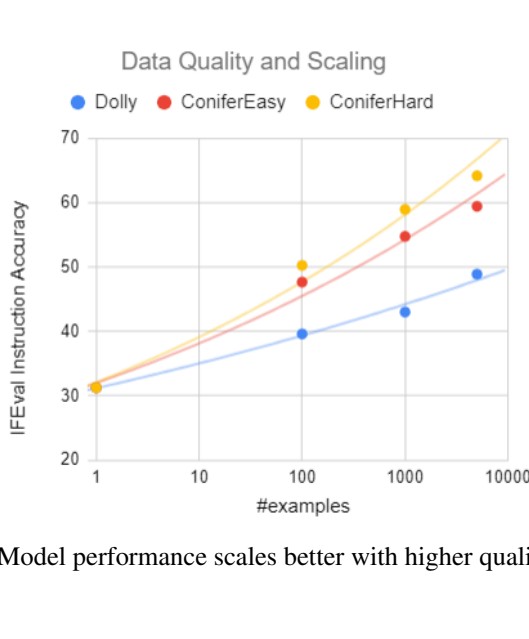

Figure 7: Model performance scales better with higher quality datasets

seen in figure 7, for the same number of examples, we see that model performance scales better with better dataset quality. This highlights the importance of high-quality datasets.

### A.3.4 FACTS100 DATASET

The prompts used for evaluating hallucination and reasoning errors in the responses for the Facts100 multihop questions are given below:

**Prompt**: You are evaluating incorrect responses to a multi-hop reasoning question. You are given a question, a response, and an event related to the question.

Return "Error" if the response hallucinates about the event in the response i.e. incorrectly recalls or uses the event and "No Error" otherwise, and nothing else.
Your job is only to check the use of the event and not the correctness of the final answer.

Now grade this:

Question - {question}
Event - {event}
response - {response}

**Prompt**: You are evaluating responses to a multihop reasoning question. You are given a question and a response.

The question is structured such that it requires a step-by-step reasoning chain to arrive at the final answer.

Your job is only to check the inaccuracy of the reasoning chain if it has one (and not of the final answer or event).
Return "Error" if the reasoning of the response is incorrect and "No Error" otherwise or if it has no reasoning, and nothing else.

Now grade this:

Question - {question}
response - {response}

## A.4 ERROR ANALYSIS PROMPT TEMPLATES

We initially collected the 100 sample results from a finetuned model and manually annotated the errors using two human reviewers to get gold-standard annotations. We then iteratively refined the prompts until the GPT-based evaluation achieved an agreement of $> 0.8$ with the human annotations.

Below are the error analysis prompts used for GSM8k evals. The placeholders inside {} are replaced by the actual question, correct answer, and the model response.

**FORMAT ANNOTATION PROMPT:**
You are given a math question, a solution, and a response. The response is supposed to be an explanation of the solution followed by the delimiter '####' and the final answer.

You are to check if the response follows this format. Return "No Error" if it follows it, and "Error" if it is not, and nothing else. It doesn't matter if the final answer is correct or not.

Now grade this:
Question: {question}
Response: {response}

**CALCULATION ANNOTATION PROMPT:**
You are given a math question, a solution, and a response. The response is supposed to be an explanation of the solution followed by the delimiter '####' and the final answer.

You are to check if the response contains any arithmetic or calculation errors. You are not required to check if the reasoning in the response is correct or not, just the arithmetic calculations.

Return "No Error" if has no calculation errors, and "Error" if it does, and nothing else.

Now grade this:
Question: {question}
Solution: {solution}
Response: {response}

**REASONING ANNOTATION PROMPT:**
You are given a math question, the solution, and an incorrect response. The response is supposed to be an explanation of the solution followed by the delimiter '####' and the final answer.

You are to check if the response contains any understanding or reasoning errors, in any of its steps. You are not required to check if the arithmetic calculations in the response are correct or not, just the reasoning.

Return "No Error" if has no reasoning or understanding errors, and "Error" if it does, and nothing else.

Now grade this:
Question: {question}
Solution: {solution}
Response: {response}

SubQA Error Analysis prompt:

**FORMAT ANNOTATION PROMPT:**
You are evaluating responses to a multihop reasoning question. You are given a question and a response. The question is a multihop reasoning question and the response is supposed to have detailed subquestion-answer style reasoning steps, followed by the final answer.

Your job is to only check if the response is answered in this format. It doesn't matter if the final answer is correct or not.

Return "Error" if it doesn't follow this format or jumps straight to the final answer and "No Error" if the response attempts to answer it step-by-step, and nothing else.

Now grade this:
Question: {question}
Response: {response}

**REASONING ANNOTATION PROMPT:**
You are grading responses to a multihop reasoning question. You are given a question, intermediate step question-answer pairs that lead to the final answer as well as a model-generated response to grade.

Check if the response correctly uses the given intermediate step question-answer to answer the question.

Return "Error" if the response has an incorrect intermediate reasoning step and "No Error", and nothing else.

Now grade this:
Question - {question}
Intermediate Step Question - {subquestion}
Intermediate Step Answer - {subanswer}
Response - {response}

## A.5    ERROR ANALYSIS RESULT SAMPLES

### A.5.1    MATH

Here is a representative example of error analysis annotation at various finetuning levels for Math

---

**Question**

Toula went to the bakery and bought various types of pastries. She bought 3 dozen donuts which cost $68 per dozen, 2 dozen mini cupcakes which cost $80 per dozen, and 6 dozen mini cheesecakes for $55 per dozen. How much was the total cost?

---

---

**Base Model**

The total cost is $313.

Explanation:

The first thing we need to do is figure out how many of each item she purchased. We can use the word "dozen" as a clue here. A dozen is equal to 12, so 3 dozens means 3 x 12 = 36 donuts, 2 dozens means 2 x 12 = 24 mini cupcakes, and 6 dozens means 6 x 12 = 72 mini cheesecakes.

Next, we multiply the number of items by their price. Donuts: 36 x 68 = $2444. Mini Cupcakes: 24 x $80 = $1920. Mini Cheesecakes: 72 x $55 = $3960.

Finally, we add all of these numbers together to get our final answer: $2444 + $1920 + $3960 = $7324.

→ Incorrect Formatting/Style
→ Incorrect Arithmetic Calculation
→ Incorrect Reasoning

---

**Base Model + GSM8k-100 examples**

How many pastries did she buy? ** The number of pastries is $<< 3*12+2*12+6*12 >>=$ 108

What was the cost of all the pastries? ** The cost of all the pastries is $<< 3 * 68/12 + 2 * 80/12 + 6 * 55/12 >>=$ $160

#### 160

→ Correct Formatting/Style
→ Incorrect Arithmetic Calculation
→ Incorrect Reasoning

---

**Base Model + GSM8k-5000 examples**

How much did the donuts cost? ** The cost of the donuts is $68x3 =$204.

How much did the mini cupcakes cost? ** The cost of the mini cupcakes is $80 x 2 = $160.

How much did the mini cheesecakes cost? ** The cost of the mini cheesecakes is $55 x 6 = $330.

What is the total cost? ** Therefore, the total cost is $204 + $160 + $330 = $694.

#### 694

→ Correct Formatting/Style
→ Correct Arithmetic Calculation
→ Correct Reasoning

---

### A.5.2  MULTIHOP QUESTION ANSWERING

Here is a representative example of error analysis annotation at various finetuning levels for Multihop Question-Answering

---

**Question**

What is the other method of extending an ice hockey game exists other than the one used in the 1932 Allan Cup?

---

**Base Model**

The only way to extend an ice hockey game is by playing overtime. In this case, the teams play five minutes of sudden death overtime. The first team that scores wins the game. If no goals are scored during overtime, then the game goes into a shootout. Each team selects three shooters who will take alternating shots on goal until there is a winner.
→ Incorrect Formatting/Style
→ Incorrect Reasoning

**Base Model + SubQA-100 examples**

What is the other method of extending an ice hockey game exists other than the one used in the 1932 Allan Cup?
Let's answer this step-by-step:
Which ice hockey game was extended using another method in 1932? − > 1932 Allan Cup.
What other method of extending an ice hockey game exists other than the one used in the 1932 Allan Cup? − > sudden-death overtime.
The final answer is sudden-death overtime.
→ Correct Formatting/Style
→ Incorrect Reasoning Steps

**Base Model + SubQA-3000 examples**

What is the other method of extending an ice hockey game exists other than the one used in the 1932 Allan Cup?
Let's answer this step-by-step:
What was the method used in the 1932 Allan Cup? − > sudden-death.
What other method of extending an ice hockey game exists other than sudden-death? − > shoot-out.
The final answer is shoot-out.
→ Correct Formatting/Style
→ Correct Reasoning

## A.6 ABLATIONS WITH SYNTHETIC DATASET

We include results for GSM8k benchmark trained on synthetic data in this section. We used the MetaMath GSM8k dataset from (Yu et al., 2023) that was generated by augmenting the human generated dataset using GPT 3.5 Turbo. It contains 200,000 training examples as opposed to the 7,500 from the original GSM8k.

From Figure 8, we see that the results roughly follow the same power-law, with flatter curve. Several works like (Xu et al., 2024) have cautioned against the use of larger models for generating high-quality synthetic data, so we used human-generated data whenever available, to prevent uncontrolled variables from affecting the experimental results.

## A.7 TRAINING AND INFERENCE PARAMETERS

No PeFT methods were used, and the learning rate was set to 1e-5 with cosine decay to 0. We found that batch size has a big effect on model performance for smaller dataset sizes. The final batch sizes we used are in table 6

The evaluation was also done in a 0-shot setting to isolate improvements gained from adding few-shot examples. Sampling was done with a temperature 0f 0.4, top-p value of 0.95, and a repetition penalty of 1.1.

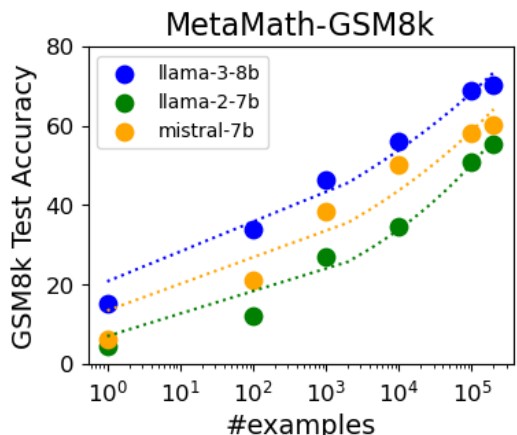

Figure 8: GSM8k performance trained on synthetic training data

Table 6: Batch sizes used.

|  | 8b & 13b models | 70b models |
|---|---|---|
| 0 - 100 | 2 | 16 |
| 101 - 1000 | 8 | 32 |
| 1001+ | 16 | 128 |

## A.8 POWER LAW FIT COEFFICIENTS

Table 7: Dataset scaling power law coefficients

| Task | Model | a | b |
|---|---|---|---|
| Math | llama-3-8b | 19.47 | 7.66 |
|  | mistral-7b | 8.14 | 4.97 |
|  | llama-2-7b | 4.21 | 5.47 |
| Multihop QnA | llama-3-8b | 11.54 | 7.77 |
|  | mistral-7b | 5.40 | 7.75 |
|  | llama-2-7b | 8.27 | 9.39 |
| Coding | llama-3-8b | 27.14 | 23.43 |
|  | mistral-7b | 17.10 | 17.47 |
|  | llama-2-7b | 3.95 | 7.11 |
| IF-Conifer | llama-3-8b | 31.06 | 11.90 |
|  | mistral-7b | 21.76 | 9.28 |
|  | llama-2-7b | 17.83 | 10.00 |
| IF-Dolly | llama-3-8b | 30.94 | 21.12 |
|  | mistral-7b | 23.76 | 15.15 |
|  | llama-2-7b | 17.04 | 14.54 |

Coefficients for the power law curves of the form $P = aD^{1/b}$ are in Figure 1 are in Table 7.

Coefficients for the power law curves in Figure 2 are in Table 8.

Table 8: Power law coefficients for Model size scaling experiment

| Model Family | Task | Model | a | b |
|---|---|---|---|---|
| llama-3 | Math | llama-3-8b | 18.18 | 7.28 |
| | | llama-3-70b | 32.07 | 9.07 |
| | Multihop QnA | llama-3-8b | 10.81 | 6.91 |
| | | llama-3-70b | 23.58 | 8.37 |
| | IF-Conifer | llama-3-8b | 33.35 | 13.52 |
| | | llama-3-70b | 37.85 | 11.83 |
| llama-2 | Math | llama-2-7b | 4.92 | 6.11 |
| | | llama-2-13b | 10.63 | 7.28 |
| | | llama-2-70b | 22.41 | 6.91 |
| | Multihop QnA | llama-2-7b | 8.37 | 9.33 |
| | | llama-2-13b | 9.83 | 6.81 |
| | | llama-2-70b | 23.09 | 9.61 |
| | IF-Conifer | llama-2-7b | 16.86 | 9.51 |
| | | llama-2-13b | 23.73 | 10.48 |
| | | llama-2-70b | 27.23 | 8.67 |

