# OpenReview forum: "Revisiting the Superficial Alignment Hypothesis"
_ICLR.cc/2025/Conference — Submitted to ICLR 2025_

### Official Review · Reviewer_ptrY · 2024-11-03

**Soundness:** 3
**Presentation:** 3
**Contribution:** 3
**Rating:** 6
**Confidence:** 4

**Summary:**

The author revisit the validity of Superficial Alignment Hypothesis made in previous work (Zhou et al., 2024). Specifically, they try to show the ground for the hypothesis is shaky because it was established on running model with chat-style data, which do not requires more knowledge. Instead, the authors choose more knowledge-intense tasks, and shows that model do learn new knowledge (beyond what's in pretraining) and such observation scales w.r.t. task data, model sizes.

**Strengths:**

* the work study the scaling factors in post-training alignment to better counter-argue superficial alignment hypothesis. This seems a novel thing that other people haven't broadly tried.
* the author basically revise the SAH to be conditional on the what it really takes to accomplish some downstream tasks. If the downstream tasks don't require more knowledge, then the alignment is only style-wise. This is a good message, although a bit obvious.
* showing that win-rate and task accuracy differs is an interesting observation.
* error analysis does seems good and comprehensive;

**Weaknesses:**

* Missing URIAL prompt [1] as a training-free alignment baseline, since the model is trained on LIMA-1k, it adds some complexity.

* The authors should also make the investigation/claim a bit more systematic or formal. In terms of capability X, how does superficial alignment hypothesis not hold (+ some analysis on error), how does the observation in capability  X different from observation in capability Y. I think the problem statement is not so well-stated.

* Some claims are not well-supported and overselling:
Line 310-314: an alternative explanation is that the model is getting there? Also, how does the overall accuracy changes? Are the number of incorrect responses decreasing? I don't think the author could conclude much here.

Line 316: the claim in Line 312-314 and here are cyclic and contradictory. The author is essentially saying "the model do align in intermediate steps/format/style, but the improvement is nothing and superficial" (Line 312-314); but, "look, we are making improvement and this tell us even more about superficial alignment hypothesis is bad". Here the same observation of "making improvement" are interpreted in two completely different standpoint, without good rationalization.

Line 370 "this post training is meant to impart reasoning ability": but does it actually improve reasoning ability? It sounds more like the authors are building their own instruction-tuned model instead of using an off-the-shelf one.

Line 404 "reasoning helps", Line 482 "these improvements are driven by the model’s reasoning ability": reasoning is a broad term here, and authors only test one kind of reasoning. thus, the claim is stronger than the supporting evidence.

* The author also didn't do thorough survey for related work. For example, [2] also study the same topic (maybe not explicitly mentioning "superficial alignment hypothesis") in a more thorough and focused manner.

[1] The Unlocking Spell on Base LLMs: Rethinking Alignment via In-Context Learning
[2] Instruction-tuned Language Models are Better Knowledge Learners

**Questions:**

* Section 3.1; could author explain why do they put heterogeneous data in the same run? Might be good to have some scaling for individual tasks, to tease apart co-learning across tasks. But this is minor and doesn't change the overall weakness.

* I note that alignment is a bit overloaded term. The purpose of the alignment could be for human-like answer, but human-like doesn't imply stronger capability. It could be interpreted that LiMA conclude the hypothesis on a specific dataset (chat-style) and should not be taken too far from its original setting?

* In Table 2 and Figure 2, instruction tuning with LiMA also seems beneficial nonetheless, where Math 10->14.7; Multihop QnA 10->21. It's just that the training instances are not as much knowledge-intense and targeted as task-1k. Doesn't this shows that the benefit of Lima data?

---

> ### Author Response · Authors · 2024-11-21
>
> We are glad that the reviewers found our work and analysis insightful. We address the weaknesses as follows:
>
> * ```...Missing URIAL prompt [1] as a training-free alignment baseline``` URIAL prompt primarily consists of two components - system prompt and K-shot In-Context Learning (ICL) over the base model. In our evaluation setting, all base and fine-tuned models included a system prompt. While evaluation was done in 0-shot, we also conducted 5-shot evaluation of math models in the lm-eval GSM8k task:
>
> | # examples (GSM8k) | system prompt + 0-shot | system prompt + 5 shot (URIAL) |
> |--------------------|----------------------|------------------------------|
> | 0 (base model)     | 10.92               | 14.27                       |
> | 100                | 37.76               | 38.29                       |
> | 1000               | 50.19               | 52.828                      |
> | 5000               | 58.68               | 61.776                      |
>
> The shape of the scaling curve remained consistent, with only an upward shift in evaluation points. This suggests URIAL prompting benefits both base and finetuned models, making it more about prompt optimization study than finetuning performance scaling.
>
> * ```...The authors should also make the investigation/claim more systematic or formal...``` We apologize for any lack of clarity. Taking mathematics as an example: the Superficial Alignment Hypothesis would suggest that mathematical ability is fully developed in the base model and would saturate GSM8k benchmark performance with just a few hundred examples. However, we demonstrate continued improvement across thousands of examples for all model families and sizes (Fig 1 and 2). Our error analysis shows that with 100 examples, models learn to mimic GSM8k response style but make numerous mistakes. Similar patterns emerge in multi-hop reasoning, where initial attempts at reasoning are incorrect, improving only with more data.
>
> * ```...Also, how does the overall accuracy change? Are the number of incorrect responses decreasing?``` Figure 3's "Total Mistakes on the test set" shows overall accuracy. Incorrect responses decrease with larger datasets, correlating more strongly with improved reasoning than improved response style.
>
> * ```...Here the same observation of "making improvement" are interpreted in two completely different standpoints, without good rationalization.``` We apologize for any unclear writing. From Figure 3, we wish to drive home the point that the users care about improvement in the final task-specific evaluation metric. From this lens, the rapid “improvement in style” followed by stagnation doesn’t translate to the “improvement in the end goal of the task”, as evidenced by the still-high “total mistakes” columns. They are mimicking the response style. However, the “improvement in reasoning” is more gradual with more data, which correlates well with the end user task. We have added specific examples of generations for further evidence in Appendix A.5 in the revised manuscript.
>
> * ```It sounds more like the authors are building their own instruction-tuned model instead of using an off-the-shelf one...``` We didn’t use an off-the-shelf-instruction-tuned model because of the reason highlighted in footnote 2. Llama-3-8b-instruct is strongly post-trained to refuse answering questions beyond their knowledge cutoff to prevent hallucination. Attempts to break this behavior by strong finetuning hampers its general capabilities.
>
> * ```...reasoning is a broad term here, and authors only test one kind of reasoning``` Since we evaluate each task independently, we define reasoning as a correct sequence of steps that helps in arriving at the final answer. In addition, we check for adjacent capabilities, for instance, arithmetic calculations in Figure 3 or hallucinations in Figure 5, and label them differently, instead of considering them as another reasoning capability.
>
> * ```...The author also didn't do a thorough survey for related work.``` We agree that the literature survey could be more thorough and we have updated the manuscript accordingly.

---

> > ### Author Response · Authors · 2024-11-21
> >
> > We address the questions as follows:
> >
> > * ```...could author explain why do they put heterogeneous data in the same run?``` We aren’t adding heterogeneous data in the same run - each task is finetuned and evaluated separately, starting from the base model, exactly like how the reviewer mentions.
> >
> > * ```...It could be interpreted that LiMA conclude the hypothesis on a specific dataset (chat-style) and should not be taken too far from its original setting?``` We agree with the reviewer’s interpretation, and this is also the point we hope to drive home from this study. We show that the claims in LIMA that “all of a model’s ability is acquired during pretraining” is an oversimplification, and doesn’t translate to complex reasoning tasks beyond simple chatbot-style stylistic alignment.
> >
> > * ```...It's just that the training instances are not as much knowledge-intense and targeted as task-1k. Doesn't this shows that the benefit of Lima data?``` While general-purpose alignment can help models better understand query intent, our study shows that task-specific supervised examples provide greater benefits within a given data budget compared to general-purpose chat-bot style alignment. LIMA and URIAL don't evaluate beyond chat-bot style QA, so we hope our comprehensive study helps inform research on data and AI alignment.

---

> > > ### Comment · Reviewer_ptrY · 2024-11-26
> > >
> > > Thank author for the intriguing discussion and additional experiments. I think my concerns are addressed. I decided to maintain my score.
> > >
> > >
> > > > we have updated the manuscript accordingly.
> > >
> > > It would be clearer if the author could highlight (e.g. color code) the added text.

---

> > > > ### Author Response · Authors · 2024-12-03
> > > >
> > > > We thank the reviewer for their comments and suggestions, which have improved the work.

---

### Official Review · Reviewer_7xz1 · 2024-11-04

**Soundness:** 2
**Presentation:** 3
**Contribution:** 2
**Rating:** 5
**Confidence:** 4

**Summary:**

This paper presents an empirical investigation to re-evaluate the superficial alignment hypothesis, which posits that the majority of a large language model's (LLM's) abilities and knowledge are acquired during pre-training. By conducting extensive experiments with diverse pre-trained models (e.g., LLaMA) and fine-tuning tasks (e.g., mathematical reasoning), and incrementally increasing the number of fine-tuning examples, the authors demonstrate that the fine-tuning scaling law applies across a wide range of capabilities. This finding challenges the original Superficial Alignment Hypothesis, which asserts that post-training does not impart new capabilities to a model.

**Strengths:**

This paper aims to revisit the superficial alignment hypothesis, which posits that post-training primarily concerns style and format adjustments, while most capabilities and knowledge are learned during pre-training.The authors undertake comprehensive experiments to explore the scaling law in post-training for various reasoning abilities, thereby illuminating a feasible pathway for enhancing the advanced capabilities of large models.

**Weaknesses:**

The superficial alignment hypothesis has not gained widespread acceptance in practical LLM pre-training. As stated in the Llama 3 technical report [1], the importance of improving performance for specific capabilities such as code/math/reasoning/tool use, etc., during post-training is emphasized. Similar conclusions have already been observed in existing technical reports or papers (e.g., the Figure 2 of paper [2]). This suggests that the re-evaluation presented in the paper may not be as novel as claimed.

Additionally, work [2] delves into how each ability varies due to the composition of supervised fine-tuning (SFT) datasets. The authors investigate this using varying amounts of mixed data and compare the results with individual ability performance, as illustrated in the Figure 3. However, the current submission only mentions multi-task settings as a potential future direction.

The distinction between pre-training and post-training has become increasingly blurred, especially with the widespread adoption of synthetic data. It is often challenging to definitively categorize synthetic question-answering pairs as either pre-training or post-training data. Moreover, there is typically an intermediary process between pre-training and post-training. Consequently, the exclusive focus on the scaling of post-training may be overly restrictive.

[1] The Llama 3 Herd of Models, https://arxiv.org/abs/2407.21783

[2] How Abilities in Large Language Models are Affected by Supervised Fine-tuning Data Composition, https://arxiv.org/pdf/2310.05492

**Questions:**

The paper would benefit from expanding the size of the post-training dataset. Given that the authors are investigating the post-training scaling law, using only the Math dataset as an example for mathematical reasoning is quite limited. It is recommended that the authors consider augmenting the dataset through the use of synthetic data or by constructing a larger, more comprehensive dataset to study the scaling law (e.g., Evol-Instruct https://arxiv.org/abs/2304.12244 and WizardMath https://arxiv.org/abs/2308.09583). This would provide a more robust and comprehensive analysis of the scaling behavior in post-training.

---

> ### Author Response · Authors · 2024-11-21
>
> * ```...in the Llama 3 technical report [1], the importance of improving performance for specific capabilities such as code/math/reasoning/tool use, etc., during post-training is emphasized. This suggests that the re-evaluation presented in the paper may not be as novel as claimed.``` We provide several novel insights including: (1) systematic scaling law treatment of performance changes across model families and sizes, (2) in-depth qualitative and quantitative analysis of model behavior at different finetuning levels, showing how generations evolve from incorrect to correct responses, and (3) a hand-curated set of new facts and multihop question-answers (to be open-sourced) with experiments on learning new knowledge alongside new capabilities. We note that our study was well-underway when the llama-3 technical report was released, and their reported large-scale post-training dataset actually supports our findings.
>
> * ```...However, the current submission only mentions multi-task settings as a potential future direction...``` While [2] presents useful analysis on post-training task data mixtures, our work focuses on establishing post-training scaling laws for LLMs and directly challenging the superficial alignment hypothesis. To limit interacting variables, we studied how specific capabilities improve through post-training by measuring performance against the number of finetuning examples. Multitask settings introduce additional variables (data mixtures, combined metrics, cross-task transfer) that were beyond our scope.
>
> * ```...Consequently, the exclusive focus on the scaling of post-training may be overly restrictive.``` While we agree this distinction isn't absolute for LLM Alignment, it remains a useful framework, as evidenced by technical reports like Llama 3 that clearly separate pre-training and post-training approaches. We define pre-training as developing general language modeling capability (evaluated via NLL loss) and post-training as improving specific human-valued tasks like math, coding, and instruction following. These stages have distinct data formats and collection methods. While high-quality pre-training remains mostly opaque within industry labs, post-training offers opportunities for controlled, observable experiments. We plan to explore other stages like mid-training or continual learning in future work.
>
> * ```...It is recommended that the authors consider augmenting the dataset through the use of synthetic data or by constructing a larger, more comprehensive dataset to study the scaling law...``` We have conducted additional experiments using synthetic data to verify our results using synthetic data. Please see our unified response above.
>
> **References:**
> * [1] Llama 3: Technical Report
> * [2] Analysis of Post-training Task Data Mixtures

---

> ### Author Response · Authors · 2024-11-27
>
> Dear Reviewer 7xz1,
>
> We sincerely thank you for your time and expertise in providing your review.
>
> Based on your valuable feedback, we have added the additional experiments and discussion that extend the results of the paper to training sets that are orders of magnitude larger using synthetic datasets. We hope that we also provided enough evidence to motivate this study as well as its impact and novelty.
>
> As the discussion period is ending, we would greatly appreciate any further inputs or comments from you, to ensure that we fully addressed your questions and weaknesses.
>
> Thank you again for your efforts.

---

> > ### Comment · Reviewer_7xz1 · 2024-11-29
> >
> > Thank you for your efforts in preparing the rebuttal. After carefully considering your response, I have decided to raise my score accordingly. However, I still have some reservations about the novelty of this work, which appears to be incremental, due to similar findings have been given in existing work as mentioned in the review.

---

> > > ### Author Response · Authors · 2024-12-03
> > >
> > > We thank the reviewer for their response and the constructive comments that have improved the paper. We understand the concern about other works highlighting similar scaling behavior during their finetuning process in narrow domains. However, we believe our work to be the first comprehensive study of this. We also added a qualitative and quantitative analysis of the development of model response behavior during posttraining. We also have a section that studies not just a model's ability to learn new capabilities but also new knowledge in posttraining, we are open-sourcing the dataset we created in that study. We believe that putting all of this together gives a very holistic report on how models evolve during posttraining, while also serving as the starting point for several new directions of research.

---

### Official Review · Reviewer_NfiD · 2024-11-04

**Soundness:** 3
**Presentation:** 3
**Contribution:** 3
**Rating:** 6
**Confidence:** 3

**Summary:**

This papre studies the superficial alignment hypothesis that  a language model's abilities and knowledge are learned during pre-training, while post-training is about giving a model the right style and format.  The authors investigate this hypothesis by assessing scaling laws in post-training across multiple tasks and model sizes, showing that fine-tuning enhances task-specific performance in reasoning and coding beyond stylistic alignment. Experiments show that while stylistic improvements are quick to saturate, task-specific reasoning, mathematical ability, and knowledge integration continue to benefit significantly with additional fine-tuning data.

**Strengths:**

1. The paper conducts a comprehensive evaluation using well-defined benchmarks,  providing evidence that post-training improves more than stylistic alignment.
2. New insights: by analyzing the power-law relationship in post-training, it shows that model improvements align well with scaling laws observed in pre-training.

**Weaknesses:**

- GPT-based Evaluation: Using GPT-based evaluation for alignment and error analysis introduces potential biases.
- Missing Citation: there are some related work that also demonstrate the scaling law in the post-training stage, sucu as this one (https://openreview.net/forum?id=MpCxUF8x61).

**Questions:**

1. How does the model's performance change when it is post-trained on datasets with diverse task formats, beyond those used here?
2. What are the underlying mechanisms that allow LLMs to integrate new knowledge effectively during post-training?
3. What are the long-term implications of extensive post-training on a model's performance across tasks? Does it risk overfitting to specific tasks or styles, potentially diminishing its generalization capabilities?

---

> ### Author Response · Authors · 2024-11-21
>
> We are glad that the reviewers found our work and analysis insightful. We address the comments and questions as follows:
>
> * ```GPT-based Evaluation```: We agree that naively using GPT-based analysis might introduce biases. Before running the GPT-based prompts on the data, we initially collected 100 sample results from a finetuned model and manually annotated the errors using two human reviewers to get gold-standard annotations. We then refined the prompts until the GPT-based evaluation achieved an agreement of $>0.8$ with the human annotations. We have described this in the updated manuscript in Appendix A.4 and added examples of model generations and their corresponding annotations.
>
> * ```Missing Citation``` Thank you for pointing this out, we have updated the manuscript with an expanded literature review.
>
> * ```How does the model's performance change when it is post-trained on datasets with diverse task formats, beyond those used here?``` This is an interesting area for future work. Our error analysis experiments used models that followed step-by-step reasoning before providing the final answer. As shown in Figure 3, formatting errors are resolved with relatively little post-training data, and while we expect this pattern to hold across diverse formats, it warrants further investigation in future work.
>
> * ```What are the underlying mechanisms that allow LLMs to integrate new knowledge effectively during post-training?``` We hypothesize that posttraining helps a model understand how to effectively utilize available knowledge, as demonstrated during finetuning on pre-cutoff data. Subsequently, it can better use new knowledge received through further finetuning/in-context learning. We verify this in our ablation study in Appendix A.2. Other works like [1] and [2] have also motivated studying the effects of post-training on new knowledge.
>
> * ```Does it risk overfitting to specific tasks or styles, potentially diminishing its generalization capabilities?``` This is a critical consideration for future work. We need better descriptions of how post-training causes catastrophic forgetting and overfitting. While the common approach is to post-train on large, diverse datasets (as shown in the llama3 technical report), more work is needed to understand the implications in the low data regime. Our work focuses on establishing that new knowledge is learned during posttraining, proposing posttraining scaling laws, and highlighting proper evaluation using relevant benchmarks. We look forward to future work extending these ideas to questions of broader, multitask generalization.
>
> **References:**
> * [1] Does Fine-Tuning LLMs on New Knowledge Encourage Hallucinations?
> * [2] Instruction-tuned Language Models are Better Knowledge Learners

---

> ### Author Response · Authors · 2024-11-27
>
> Dear Reviewer NfiD,
>
> We sincerely thank you for your time and expertise in providing your review.
>
> Based on your valuable feedback, we have added a new section that details the rigorous treatment that GPT-based evaluation underwent, to prevent biases. We have also provided example generations that illustrate the annotations from the evaluation.
>
> As the discussion period is ending, we would greatly appreciate any further inputs or comments from you, to ensure that we fully addressed your questions and weaknesses.
>
> Thank you again for your efforts.

---

### Official Review · Reviewer_pyTF · 2024-11-04

**Soundness:** 3
**Presentation:** 2
**Contribution:** 2
**Rating:** 6
**Confidence:** 4

**Summary:**

This work challenges the superficial alignment hypothesis, which is defined by three rules:  1) A model's knowledge is learned entirely during pre-training. 2) A small number of examples can saturate a model's performance for a given task. 3) Post-training is largely about style and doesn't does not teach a model new capabilities. This work illustrates a post-training scaling law and offers several insights, claiming that the model can learn new skills during the post-training and it also requires much more training data.

**Strengths:**

This work presents in-depth analyses, offering insights for understanding the post-training process. It finds that the format-related errors are often saturated with a few samples, but the reasoning ability requries much more examples to improve.

**Weaknesses:**

Since this work aims to study the scaling law of post-training, the choice of task is limited by the size of the training set. However, the training set is still small for building a reliable scaling law. One solution is to verify the idea of synthetic tasks.

Except for several insights of analyzing the detailed post-training process, the main findings (the key takeaways section) are not a surprise; some of them have been discussed in the community. The contribution of this work is more like to be a verficiation and provide more evidence. Besides, this work uses the term "post-training" frequently, but the actual experiments are limited to SFT rather than RLHF.

**Questions:**

see weakness
typos， the C3 and C2 at L055 are in the wrong order.

---

> ### Author Response · Authors · 2024-11-21
>
> We are glad that the reviewers found our work and analysis insightful. We address the comments and questions as follows:
>
> * ```However, the training set is still small for building a reliable scaling law, One solution is to verify the idea of synthetic tasks.``` We have conducted additional experiments using synthetic data to verify this. Please see our unified response above.
>
> * ```...The contribution of this work is more like to be a verification and provide more evidence.``` We agree that some of the key insights are intuitive, but a rigorous treatment of this behavior is either conducted in closed industry research or is anecdotal evidence. As the reviewer pointed out, we have done several in-depth analyses in terms of model behavior at various scales, for different tasks, at different model sizes and dataset scales, using open-source datasets and models. This would be valuable information for the vast majority of researchers who intend to collect data for their use cases, as well as smaller industry teams that want to improve LLMs for a particular task. We also provide a detailed study of learning new knowledge during post-training with novel insights on evaluation and hallucination.
>
> * ```...this work uses the term 'post-training' frequently, but the actual experiments are limited to SFT``` Please refer to the unified response to this comment above.
>
> * ```...the C3 and C2 at L055 are in the wrong order.``` Thank you for pointing this out, we have fixed this.

---

> ### Author Response · Authors · 2024-11-27
>
> Dear Reviewer pyTF,
>
> We sincerely thank you for your time and expertise in providing your review.
>
> Based on your valuable feedback, we have added the additional experiments and discussion that extend the results of the paper to training sets that are orders of magnitude larger using synthetic datasets. We have also improved the writing of the paper to clarify the use of our terms.
>
> As the discussion period is ending, we would greatly appreciate any further inputs or comments from you, to ensure that we fully addressed your concerns and weaknesses.
>
> Thank you again for your efforts.

---

> > ### Comment · Reviewer_pyTF · 2024-12-02
> >
> > Thanks the authors for the detailed responses. I would like to keep my scores.

---

> ### Author Response · Authors · 2024-12-03
>
> We thank the reviewer for their comments and suggestions, which have improved the work.

---

### Author Response · Authors · 2024-11-21
**Unified response**

1. **Regarding comments about our use of the terms 'Posttraining' and 'SFT':** Our terminology follows the precedent set by LIMA [1], which established SFT as a key method for studying posttraining and alignment. While our study specifically examines supervised finetuning as a posttraining method, we acknowledge the broader scope of alignment techniques and have designated RLHF exploration for future work. We have clarified this scope in the manuscript and acknowledged this in the discussion of limitations and future directions.

2. **We specifically chose to use high-quality human-generated or verified datasets** instead of synthetic datasets to ensure experimental control. Synthetic data also introduces a ceiling on the quality of the generating model in terms of performance, which might affect the experimental study [2]. However, we understand that the size of such high-quality datasets is limited, and **we have added additional experiments using augmented data in the appendix.** The scaling pattern holds even on high-quality synthetic data, for much larger order-of-magnitude dataset sizes.

3. **We have updated the manuscript** with example generations of models at various fine-tuning levels and error annotations that qualitatively show how models improve in style vs. reasoning.

**References:**
* [1] LIMA: Less Is More for Alignment
* [2] Stronger Models are NOT Stronger Teachers for Instruction Tuning

---

### Meta-Review · Area_Chair_GTbc · 2024-12-20

**Metareview:**

This paper revisits and provides evidence against the superficial alignment hypothesis from the LIMA paper, which states that only a few samples are needed to "align" a language model, and that this alignment is only superficial. The paper finds that this is not really the case, and more substantial post-training alignment can change the model in substantial ways.

On the plus side, this paper is well executed, and the empirical results seem thorough (if somewhat too small to establish scaling laws with). On the negative side, the superficial alignment hypothesis itself was not really accepted as an established finding in the community, and thus the results aren't really surprising (my sense is that the null hypothesis in the community is still that post-training alignment changes the model capabilities substantially, and the LIMA paper, while interesting, did not show enough evidence to reject the null hypothesis).

Therefore, I am recommending that this paper not be accepted.

**Additional Comments On Reviewer Discussion:**

Reviewer NfiD gave a somewhat high score of 6 but seemed to be a low effort review, so this was discounted. Reviewer ptrY and Reviewer pyTF did not change their scores after engaging with the authors during the discussion period. While Reviewer 7xz1 raised their score after rebuttal, it was still a borderline score. These factors resulted in my recommendation to reject the submission.

---

### Decision · Program_Chairs · 2025-01-22

Reject